# STaR-GATE:
# Teaching Language Models to Ask Clarifying Questions

**Chinmaya Andukuri**\*
Stanford University
andukuri@stanford.edu

**Jan-Philipp Fränken**\*
Stanford University
jphilipp@stanford.edu

**Tobias Gerstenberg**
Stanford University
gerstenberg@stanford.edu

**Noah D. Goodman**
Stanford University
ngoodman@stanford.edu

## Abstract

When prompting language models to complete a task, users often leave important aspects unsaid. While asking questions could resolve this ambiguity (GATE; Li et al., 2023), models often struggle to ask good questions. We explore a language model's ability to self-improve (STaR; Zelikman et al., 2022) by rewarding the model for generating useful questions—a simple method we dub STaR-GATE. We generate a synthetic dataset of 25,500 unique persona-task prompts to simulate conversations between a pretrained language model—the `Questioner`—and a `Roleplayer` whose preferences are unknown to the `Questioner`. By asking questions, the `Questioner` elicits preferences from the `Roleplayer`. The `Questioner` is iteratively finetuned on questions that increase the probability of high-quality responses to the task, which are generated by an `Oracle` with access to the `Roleplayer`'s latent preferences. After two iterations of self-improvement, the `Questioner` asks better questions, allowing it to generate responses that are preferred over responses from the initial model on **72%** of tasks. Our results indicate that teaching a language model to ask better questions leads to better personalized responses.

## 1 Introduction

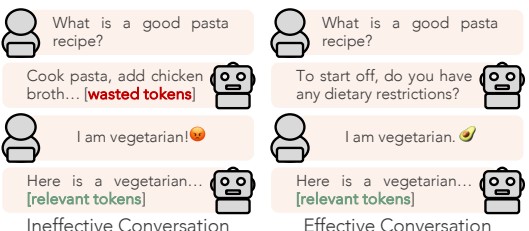

Figure 1: **Problem Illustration**. When user preferences are unknown, language models may respond ineffectively. By asking questions, models can elicit information and provide more effective responses.

When interacting with users who have different preferences, language models (LMs) encounter *task ambiguity* (Finn et al., 2018; Tamkin et al., 2022). Depending on the user, the same request might correspond to a different task. For example, consider a user who asks an LM for a pasta recipe (Figure 1). If the model could elicit information about the user's dietary restrictions, favorite sauces, and preferred cooking methods, it could tailor the recipe to their specific needs and desires. The model might suggest a vegetarian pasta recipe for a user who is vegetarian, or propose a traditional lasagna recipe for a user with a passion for Neapolitan cuisine. However, if this information is not explicitly specified in the prompt, the model may generate a generic recipe that fails to account for the user's unique preferences and constraints. In high-stakes domains like healthcare or education, such task ambiguity can have significant consequences.

One approach to resolving task ambiguity is by asking targeted questions to elicit relevant information from users. Prompting closed-source LMs can yield useful questions (e.g., Li

---

\*Equal contribution.

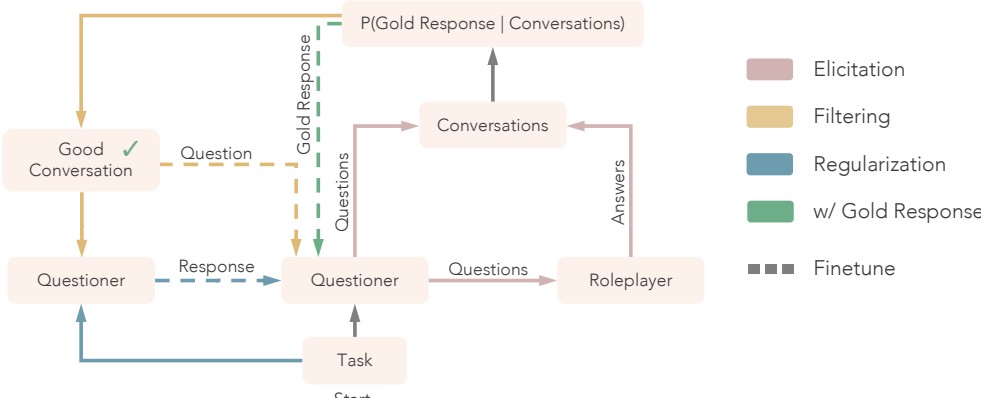

Figure 2: **Overview of STaR-GATE**. A task is given to a `Questioner` who **elicits** preferences from a `Roleplayer` whose persona is unknown to the `Questioner`. The resulting conversations are then **filtered** based on the log probability of a gold response generated by an `Oracle` which has access to the `Roleplayer`'s persona (omitted from the diagram for clarity). We then fine-tune the `Questioner` on the filtered questions. Moreover—to avoid distribution shift—we **regularize** the `Questioner` by additionally sampling responses conditioned on the filtered conversations. In our ablations, we contrast fine-tuning on sampled responses with fine-tuning on the **gold responses**.

et al., 2023; Piriyakulkij et al., 2023). However, this approach is inflexible in guiding a model's questioning strategy and frequently generates queries that are ineffective or irrelevant for the task at hand. Indeed, it is likely that current alignment strategies—such as RLHF—specifically inhibit the ability to carry out such dialog (Shaikh et al., 2023). One recent effort addresses these limitations by combining elicitation with optimal experimental design methods (Handa et al., 2024). However, this approach constrains questions to pairwise comparisons over a fixed set of features, substantially limiting the space of questions that can be used to probe user preferences. Another approach is to use offline reinforcement learning to encourage useful dialog (Hong et al., 2023). This is promising but requires offline generation of high-quality dialog from an expert model, and has not targeted questions for preference elicitation specifically.

In this paper, we explore whether we can improve a LM's ability to ask useful questions by bootstrapping with a form of self-play (Silver et al., 2017; Anthony et al., 2017). We introduce **STaR-GATE** (Figure 2), an iterative algorithm that combines active preference elicitation (GATE; Li et al., 2023) with a self-improvement loop inspired by STaR (Zelikman et al., 2022). We address several technical challenges: (1) We define a task setting for improving elicitation for which we generate a **synthetic dataset** of 25,500 unique persona-task prompts; (2) We define a **reward function** based on the log probability of gold responses generated by an oracle model (with access to the persona); and (3) We encourage the LM to use the elicited information while avoiding distribution shift through **response regularization**. We find that questions asked by the finetuned model increase the probability of gold responses consistently across iterations (Figure 4). Moreover, compared to responses generated by the initial model, responses generated by a STaR-GATE finetuned model have **72%** win rates (Figure 3a) after two iterations.

In **summary**, we make the following **contributions**: **(1)** We introduce STaR-GATE, a simple algorithm that iteratively improves a LM's ability to elicit user preferences by asking questions. **(2)** We generate a synthetic dataset consisting of 25,500 unique persona-task-response prompts. **(3)** We show that finetuning with STaR-GATE enables a LM to generate questions that significantly increase the probability of generating gold responses. **(4)** We show that adding response-regularization to STaR-GATE yields a fine-tuned model able to use the elicited preferences to generate better responses—a high win rate against the initial model. **(5)** We show that the finetuned model generalizes beyond the roleplayer it was trained with.

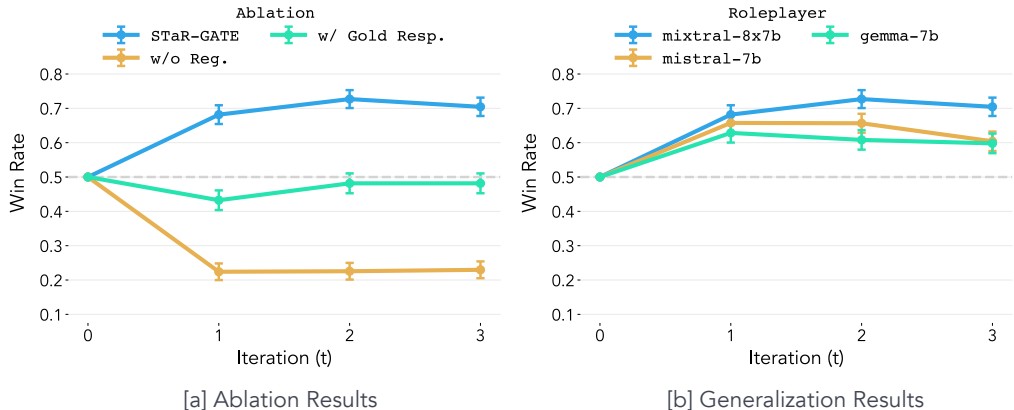

Figure 3: **Win Rates Against Initial Model.** [a] Complete method and ablations: `w/o Reg.` refers to finetuning on questions only, which decreases the model's ability to generate answers. `w/ Gold Resp.` refers to finetuning directly on the gold responses rather than self-generated responses, which leads to hallucinations in generated answers. [b] `Roleplayer` generalization results. We demonstrate that STaR-GATE generalizes beyond the roleplayer it was trained against (`mixtral-8x7b`). All three roleplayers correspond to the instruct version of their respective models. Error bars represent the standard error of the mean ($\pm$ SEM). We include 0.5 (chance) as a reference point for iteration $t = 0$.

## 2   Related Work

**Preference Optimization.** Preference optimization algorithms, such as RLHF (Christiano et al., 2017), DPO (Rafailov et al., 2024), or KTO (Ethayarajh et al., 2024), optimize LMs to provide single-turn dialog responses that reflect preferred or high-utility outcomes. As a result, these models learn distributions over responses that effectively generate answers to user queries without requiring additional information beyond the initial prompt. However, asking follow-up questions to elicit user preferences is essential for understanding their unique needs and desires, especially when faced with task ambiguity (Tamkin et al., 2022; Li et al., 2023). Despite the importance of follow-up questions for effective communication, recent research has shown that preference optimization algorithms can **reduce** a LM's ability to ask follow-up questions. Specifically, RLHF has been found to negatively correlate with a LM's attempts to ask follow-up questions or show acknowledgment (Shaikh et al., 2023). This limitation can be problematic for high-stakes domains such as healthcare (Thirunavukarasu et al., 2023) or education (Kasneci et al., 2023) , where resolving task ambiguity through effective questioning is crucial for effective dialog.

**Preference Elicitation with LMs**. One way of resolving task ambiguity is by prompting a LM to ask questions or infer user preferences from observations (Li et al., 2023; Piriyakulkij et al., 2023; Lin et al., 2023; Fränken et al., 2023; Handa et al., 2024; Rao & Daumé III, 2018; 2019; Aliannejadi et al., 2021; Mostafazadeh et al., 2017). For example, Li et al. (2023) used LMs *themselves* to elicit user preferences during interaction (**GATE**; Li et al., 2023). In GATE (short for Generative Active Task Elicitation), a LM elicits and infers intended behavior through free-form, language-based interaction. Unlike non-interactive elicitation approaches, such as prompting (Brown et al., 2020), which rely entirely on the user to specify their preferences, generative elicitation probes nuanced user preferences better. Across domains such as content recommendation and email verification, generative elicitation with LMs requires less effort than prompting while being comparable to or better than user-written prompts (for further details, see Section 5 in Li et al., 2023). Building upon GATE, Handa et al. (2024) introduced OPEN, a framework that combines LM-driven elicitation with Bayesian Optimal Experimental Design (BOED) to select informative questions and translate abstract queries into natural language. OPEN combines the advantages of LMs and Bayesian methods to recommend news articles. OPEN is better at eliciting human preferences than approaches that only use LMs or BOED. Piriyakulkij et al. (2023) combine

---

**Algorithm 1** STaR-GATE

---

1: Input $Q_{BASE}$: a pretrained LM; tasks $T = \{(t_i)\}_{i=1}^{D}$, personas $U = \{(u_j)\}_{j=1}^{C}$, and gold responses $G = g_{ij}$ for $ij \in\in \{1, ..., D\} \times \{1, ..., C\}$
2: **for** $\eta$ in $1...N$ **do**                                                                    ▷ Outer loop
3:     $\{s_{ij}^c\}$ with $c \in [1, 10] \leftarrow Q_{\eta-1}(t_i, u_j)$ for $ij \in \{1, ..., D\} \times \{1, ..., C\}$          ▷ Simulate multiple conversations for each $(ij)$
4:     $S_\eta = \{s_{ij}^*\} \leftarrow \arg\max_{s_{ij}^c} \log p_{Q_{BASE}}(g_{ij}|t_i, s_{ij}^c)$ with $c \in [1, 10]$ for $ij \in \{1, ..., D\} \times \{1, ..., C\}$    ▷ Filter conversations with highest log-probabilities of generating the gold responses according to the original model $Q_{BASE}$
5:     $R_\eta = \{Q_{\eta-1}(t_i, s_{ij}^*)\}$    ▷ Generate model responses conditioned on the best selected conversations for all $ij$ using $Q_{\eta-1}$
6:     $Q_\eta \leftarrow \text{train}(Q_{BASE}, S_\eta, R_\eta)$ ▷ Finetune the original model on selected conversations and model responses
7: **end for**

---

LMs with probabilistic reasoning to select informative questions that maximize information gain about user preferences in a simplified web shopping task (Yao et al., 2022). Relatedly, Hong et al. (2023) demonstrated that instead of eliciting information directly, it is also possible to prompt a large LM such as GPT-3.5 to simulate conversations between a human and an assistant, and then revise the simulated conversation with Constitutional AI (Bai et al., 2022). This approach allowed the authors to finetune a much smaller GPT-2 model (Radford et al., 2019) to become a capable conversationalist. While all of the above approaches have resulted in significant improvements, they rely on proprietary models for both elicitation and generation of synthetic data for downstream finetuning.

**Self-Improving Reasoning**. We are interested in *training* a LM to better elicit preferences using its *own* reasoning capabilities. To do so, we draw upon recent work showing that LMs can *self-improve*. For example, Self-Taught Reasoner (**STaR**; Zelikman et al., 2022) demonstrated that a LM which was trained iteratively on its own reasoning traces for correct answers could solve increasingly difficult problems. By combining rationalization (i.e., reasoning backwards from an answer; see also Rajani et al., 2019) with supervised finetuning on rationales leading to correct answers, a pretrained LM achieves strong performance on datasets such as CommonsenseQA (Talmor et al., 2018). Recently, V-STaR (Hosseini et al., 2024) extended this idea by using both correct *and* incorrect reasoning traces, essentially attempting to merge STaR with DPO. Relatedly, TRICE (Hoffman et al., 2024) frames the process of generating better chains of thought as a latent-variable inference problem and maximizes the marginal log-likelihood of correct answers. Other relevant works include learning intermediate reasoning for mathematical statements (Poesia et al., 2023), learning from reasoning mistakes (Shinn et al., 2024; Zhang et al., 2024), learning to follow constitutions (Fränken et al., 2024), teaching LMs to reason in planning (Gandhi et al., 2023; Qiao et al., 2024), and Quiet-STaR (Zelikman et al., 2024), a generalization of STaR which generates rationales at each token to explain future text. Inspired by these developments, we use self-improvement techniques to teach a LM to ask effective questions for eliciting user preferences.

## 3 STaR-GATE

**Overview.** On a high level, STaR-GATE starts with persona-task prompts and generates **gold responses** with an `Oracle` that has access to both the persona and the task. Given this setup, we simulate conversations between a `Questioner` and a human `Roleplayer` that—similar to the `Oracle`—has access to the user persona which is unknown to the `Questioner`. The task of the `Questioner` is to elicit useful information from the Roleplayer, whereby usefulness is measured as the log probability of the gold response conditional on the questions asked by the `Questioner` and the preferences elicited from the `Roleplayer` (see Figure 2).

**Objective.** Let $Q$ denote the Questioner (i.e., the **policy** to train), $R$ the Roleplayer model, $T$ the set of tasks, $U$ the set of user personas, and $O$ the Oracle model. Given a task $t_i \in T$ and a persona $u_j \in U$, the Oracle $O$ generates a gold response $g_{ij} \sim p_O(g|t_i, u_j)$. The objective of STaR-GATE is to maximize the expected log probability that the *pretrained* model $Q_{BASE}$ assigns to the gold response $g_{ij}$, given the task $t_i$ and a simulated conversation $s_{ij}$ between $Q$ and $R$:

$$J(Q, R, T, U) = \sum_{t=i}^{T} \sum_{u=j}^{U} \mathbb{E}_{s_{ij}} \log p_{Q_{BASE}}(g_{ij}|t_i, s_{ij}). \tag{1}$$

Here $s_{ij} := [q_{ij1}, h_{ij1}, \ldots, q_{ijk}, h_{ijk}]$ is a simulated conversation of questions $q_{ijk}$ distributed according to $p_Q(q_{ijk}|t_i, q_{ij1}, h_{ij1}, \ldots, q_{ijk-1}, h_{ijk-1})$ and answers distributed according to $p_R(h_{ijk}|u_j, t_i, q_{ij1}, h_{ij1}, \ldots, q_{ijk})$.

**Optimization.** Equation 1 can be optimized in a variety of ways. Following Zelikman et al. (2022), we use a simple variant of Expert Iteration (Anthony et al., 2017). On each overall iteration, $\eta$, for each pair $(t_i, u_j)$, we sample $N$ trajectories of simulated conversations, $s_{ijn}$, using the current $Q_\eta$. We then select the top-$k$ trajectories (here, $k = 1$) based on the objective, and do supervised fine-tuning for this set from the initial $Q_{BASE}$.

**Regularization.** An important failure mode of optimizing this objective is that by training the policy $Q$ to ask good questions it may forget how to respond and instead always ask questions. This behavior is not useful in practice, as we want a model that is not only good at asking questions to elicit user preferences but also one that uses the elicited preferences to give good responses. To address this issue, we add to Equation 1 a regularization term preventing the distribution of *responses* (not questions) from moving too far from the previous iteration: $KL(p_{Q_{\eta-1}}(r|t_i, s_{ij})||p_{Q_\eta}(r|t_i, s_{ij}))$. In practice this can be accomplished by simply sampling a response (at temperature $\tau = 0$) from the previous policy $r_{ij} \sim p_{Q_{\eta-1}}(r|t_i, s_{ij}^*)$ for each task and persona pair, conditioned on the best conversation history $s_{ij}^*$, then appending this response to the conversation history during fine-tuning.

**Algorithm.** We provide an outline of STaR-GATE in Algorithm 1. We perform expert iteration, training the initial model $Q_{BASE}$ $\eta$ times ($N = 3$ for all experiments) on question-response pairs generated from each intermediate model $Q_1, Q_2, ..., Q_\eta$. At each iteration $\eta$, we alternate between task splits $T_A$ and $T_B$, as well as persona splits $U_A$ and $U_B$, to prevent generating new data for tasks or personas present during training. For each task $t_i$ and user persona $u_j$, we simulate $n$ conversations ($N = 10$ for all experiments), each having a maximum of $K$ total turns ($K = 3$ for all experiments). When generating $s_{ij}$ for all $(ij)$, we sample $(q_{ijk}, h_{ijk})$ at each turn $k$ from the previous $Q_{\eta-1}$ and fixed $R$. To achieve a roughly uniform distribution of conversation lengths and prevent overfitting on conversations of a single length, we set the termination point to be uniform across $K$.

As indicated above, we select the best simulated conversations for finetuning the next iteration according to the objective $p_{Q_{BASE}}(g_{ij}|t_i, s_{ij})$. We then fine-tune the initial model $Q_{BASE}$ on both the selected conversations $s_{ij}^*$ and the greedily sampled responses $r_{ij}$, ensuring that the model learns to ask informative questions *and* provide personalized responses. Critically, we mask the answers $h$ from the loss, finetuning the question-generation and response-generation policy but not learning to imitate answers.

## 4 Elicitation Task

**Overview.** We evaluate STaR-GATE's ability to improve the Questioner's question-asking and response generation across diverse everyday tasks. We find that training with STaR-GATE increases both the log-probability of gold responses and win rates compared to the initial (pretrained and instruction-finetuned) model. Code to reproduce experiments is available at https://github.com/scandukuri/assistant-gate.

To cover a broad range of everyday life tasks, we selected the first 550 conversations from the open-source `instruct-human-assistant-prompt-dataset`[1] which we divided into two train splits ($T_A$, $T_B$) each of $N = 250$ and one test split $N = 50$. Importantly, we only selected the human queries (not the assistant responses) for each conversation and used these as the tasks $t \in T$ to seed a given simulation. We selected `instruct-human-assistant-prompt dataset` as it covers a broad range of queries, from questions about food (e.g., *"What type of wine goes best with steak?"*), to career questions (e.g., *"I'm having trouble finding the perfect job. What resources can help me?"*), and education (e.g., *"I'm curious about quantum computing. Can you tell me the basics of how it works."*). See Appendix A.2 for further details.

**Persona Generation.** We generate personas $u \in U$ with GPT-4 by few-shot ($N = 2$) prompting with randomly sampled personas from a set of 21 content-filtered personas[2] from the PRODIGy dataset (Occhipinti et al., 2023). We generated a total of 110 personas and split personas into two train splits ($U_A$, $U_B$) each with 50 personas, and one test split of $N = 10$. Example personas and prompts are provided in Appendix A.3.

**Gold Responses.** To generate a gold response $g_{ij}$ for each $(t_i, u_j)$ pair, we prompted an `Oracle` (GPT-4). Specifically, we provided the persona $u_j$ followed by the task $t_i$, without any dialog history, and prompted the `Oracle` to generate a personalized response that completes the task with respect to the persona profile. This process resulted in a total of $25,500$ task–persona–gold-response triples (250 x 50 + 250 x 50 + 50 x 10). Prompt details and examples are provided in Appendix A.4.

## 5 Evaluation and Results

We evaluate the performance of the `Questioner` $Q$ at each iteration $\eta$ using two metrics: **log-probabilities** of generating the gold responses and **win rates**.

**Models.** We use `mistral-7b-instruct` as our `Questioner`. We chose `mistral-7b-instruct`, a 7B-parameter model, because its weights are openly available and it has been shown to be one of the best models for its size (Jiang et al., 2023), outperforming larger models such as `llama-13B` (Touvron et al., 2023) on benchmarks like MT-Bench (Zheng et al., 2024). We use GPT-4 (OpenAI, 2023, `gpt-4-0613` snapshot) as our `Oracle`, as at the time of generating our dataset, GPT-4 was the most capable model available. For the `Roleplayer`, we use `mixtral-8x7b-instruct` (Jiang et al., 2024).

**Gold Log-probabilities.** Our main training objective is learning to elicit information that increases the log probability of the gold responses according to the *initial* (pretrained and instruction-finetuned) model $Q_{BASE}$. For our evaluations, we thus first compute the log probability of gold responses $g_{ij}$, conditioned on simulated conversations $s_{ij}^{(n)}$ generated by the current model $Q_\eta$ and fixed roleplayer $R$, for a held-out test set of tasks $t_i$ and personas $u_j$. We calculate log probabilities for **four conditions**: (1) **Negative Control**: $\log p_{Q_{BASE}}(g_{ij}|t_i)$, performance of the pretrained model without any information about the `Roleplayer` (persona or elicited), (2) **Positive Control**: $\log p_{Q_{BASE}}(g_{ij}|u_j, t_i)$, performance of the pretrained model given oracular information about the persona, (3) **Q-Experimental**: $\log p_{Q_{BASE}}(g_{ij}|t_i, s_{ij}^{(n)})$, evaluation of the STaR-GATE finetuned model, and (4) **Q-Random**: $\log p_{Q_{BASE}}(g_{ij}|t_i, s_{i,r \neq j}^{(n)})$, a baseline that randomizes persona info used in answering elicitation questions ($r \neq j$ indicates a random different test persona). In prompting both Q-Random and Q-Experimental (the main condition), we repeat the task text $t_i$ at the end of the conversation to prompt a final response instead of asking another elicitation question. The purpose of the Q-Random baseline is to isolate the relevance of persona-specific information from generally informative information elicited from the `Roleplayer`.

---

[1] instruct-human-assistant-prompt

[2] We hand-selected personas that did not contain references to violence, profanity, or content violations.

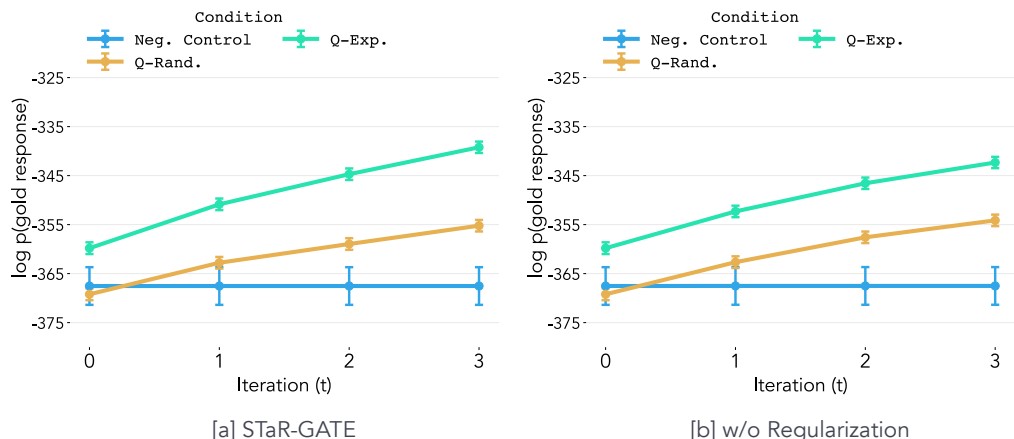

Figure 4: **Log Probability of Gold Responses**. Log probabilities of gold responses increase over iterations for both [a] STaR-GATE and [b] STaR-GATE w/o Regularization. Error bars correspond to $\pm$ SEM calculated across held-out persona-task prompts.

Our **results** show that log probabilities of the gold response **increase** over iterations for the Q-Experimental condition (Figure 4a). We observe a similar trend (however, with much lower log probabilities) for the Q-Random baseline. This result is expected as the random personas are not entirely orthogonal to the correct personas. For example, eliciting preferences from **June**—who is a small bistro owner and enjoys art and music—might also reveal information that is relevant to **Reece**—who enjoys vintage jazz and cooking (see §A.3). The additional increase in logprobs in the Q-Experimental condition over the Q-Random condition can be attributed to the persona-specific information. For the Q-Experimental/Random conditions, each data point for the log probabilities is calculated using 10 simulated conversations for each of the 10 x 50 persona-task prompts, resulting in a total of 5000 responses (which is why the error bars are small). See §A.1 for figures including log probabilities for the positive control condition.

**Win Rates**. The primary goal of asking questions is to generate high-quality answers, not just to assign high probability to known, good answers. To evaluate this, we compared the responses from the STaR-GATE model, $Q_\eta$, to those from the initial model, $Q_{BASE}$. For each $(t_i, u_j)$ pair, we used model $Q_\eta$ to generate a response $r_{ij}^{(n)}$ at temperature $\tau = 0$, conditioned on a randomly sampled conversation $s_{ij}$ at temperature $\tau = 0.9$. We then prompted GPT-4 to choose the more suitable response for task $t_i$ and persona $u_j$, following the evaluation protocol of Rafailov et al. (2024). Specifically, GPT-4 was asked to select between $r_{ij}^{(n)}$ and $r_{ij}^{(0)}$ (see Figure 16). To mitigate order effects (Wang et al., 2023), we randomized the order of the responses. Due to the uniform sampling of turn lengths, each turn length has approximately 166 $(ij)$ pairs in total. Consequently, each data point for the win rates is an average of 300 values.[3] Our **results** show that win rates for STaR-GATE **increase** over iterations (see Figure 3b), reaching a maximum win rate of **72%** after two iterations.

## 6 Ablations

We perform several ablations to study the effect of different design choices on the performance of STaR-GATE.

**Roleplayer Robustness.** To investigate the effect of `Roleplayer` capability on the `Questioner`'s performance, we conducted evaluations with different `Roleplayer` models: `mistral-8x7b-instruct`, `mistral-7b-instruct`, and `gemma-7b-instruct` (Mesnard et al.,

---

[3]GPT-4's content filter rejected 1-2% of requests, resulting in average values between 293 and 300 for each data point.

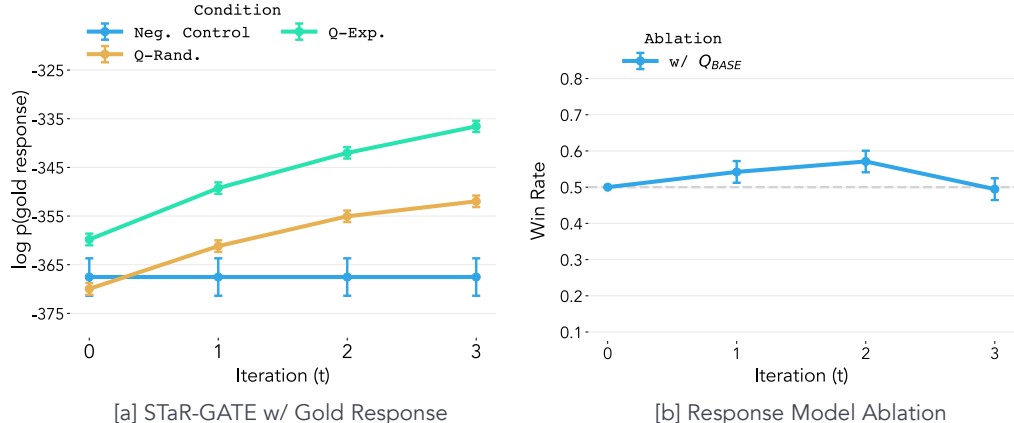

[a] STaR-GATE w/ Gold Response          [b] Response Model Ablation

Figure 5: **Additional Ablation Results**. [a] Log probability of gold responses for STaR-GATE w/ gold response. [b] Win Rates for STaR-GATE using $Q_{BASE}$ to generate responses at each iteration. Error bars correspond to $\pm$ SEM.

2024). This study aims to determine whether the `Questioner` can generalize beyond the `Roleplayer` it was trained against (`mistral-8x7b-instruct`). The **robustness** results show that when using `mistral-7b-instruct` as the `Roleplayer`, STaR-GATE achieves a slightly lower win rate of **65%** after two iterations. When `gemma-7b-instruct` is used as the `Roleplayer`, the win rates peak at **62%** after one iteration. This result shows that STaR-GATE can **generalize** to different `Roleplayers`, though with slightly lower performance (see Figure 3b).

**Training Ablations.** To demonstrate the importance of **regularization** during **training** (i.e., sampling responses $r_{ij}$ and finetuning on these; for details see Algorithm 1), we additionally run an ablation in which we only finetune on questions $q_{ij}^*$ but not responses $r_{ij}$ (see Figure 2, for an illustration). We expect that this ablation (**STaR-GATE w/o Regularization**) decreases win rates, as the `Questioner` $Q$ might forget how to respond and instead always asks questions. Finally, we include an ablation in which we finetune on the gold responses $g_{ij}$ instead of the sampled responses $r_{ij}$. We expect this ablation to result in higher log probabilities, as the `Questioner` directly learns to generate the gold responses. However, we also expect this to lead to hallucination during the generation of responses, as the `Questioner` will have seen information from gold responses that was not present in the elicited preferences (since the gold responses come from an `Oracle` that sees the complete persona). We denote this ablation as **STaR-GATE w/ Gold Response**. **Win rates** for both STaR-GATE w/o Regularization and STaR-GATE w/ Gold Response are shown in Figure Figure 3a. As expected, STaR-GATE w/o Regularization **decreases** win rates over iterations, as finetuning on questions alone yields a model that forgets how to respond (see §A.5, for an example). For STaR-GATE w/ Gold Response, win rates initially decrease and then converge to **50%**. We attribute this to hallucination in responses that were not aligned with the elicited responses (see §A.5). Log probabilities for STaR-GATE w/o Regularization are slightly lower compared to STaR-GATE (Figure 4b), while log probabilities for STaR-GATE w/ Gold Response were slightly higher (Figure 5a).

**Response Model.** We finally run an additional **win rate** evaluation in which we report GPT-4 win rates for responses generated by model $Q_{BASE}$ conditional on conversation elicited from model $Q_\eta$ over responses generated by the initial model $Q_{BASE}$ conditional on information elicited by $Q_{BASE}$ (see Figure 5b). The purpose of this condition was to understand whether the initial model would benefit from the conversation history in the same way the STaR-GATE finetuned model would. While we found a slight increase in win rates up to **57%** after two iterations, win rates eventually reversed to 50% at iteration three. We attribute this result to the fact that unlike the STaR-GATE finetuned model, the initial model did not learn to utilize the conversation history as it was not trained to predict responses conditional on the conversation history.

**Self-Oracle.** A key limitation of our previous experiments is the reliance on a more capable model (GPT-4) for oracular information. To address this, we finally conducted an exploratory analysis using `llama3-8b-instruct` for all STaR-GATE components (`Oracle`, `Questioner`, and `Roleplayer`). After one iteration of finetuning, this setup achieved a **65%** win rate against the initial model. For comparison, using `llama3-8b-instruct` as `Questioner` and `Roleplayer` with GPT-4 as `Oracle` yielded a **66%** win rate. These additional results demonstrate that if the `Questioner` is strong enough, STaR-GATE might function effectively without a significantly more powerful oracle model like GPT-4. Moreover, it suggests that a strong `Questioner` like `llama3-8b-instruct` can benefit from interacting with a `Roleplayer` of equal capabilities.

## 7 Limitations and Future Work

One important limitation of our work is its dependence on gold responses (i.e., labels). While our current approach cannot be framed as full self-play or self-improvement, using a stronger model for the `Questioner` (e.g., `llama3-8b-instruct` or `mixtral-8x7b-instruct`) could potentially enable it to function as a self-oracle, thereby eliminating the need for external gold responses. Our exploratory analysis using `llama3-8b-instruct` as a self-oracle have shown promising results in this direction. In addition to filtering based on gold responses, another extension could focus on directly supervising the questions, which might help the model ask even more effective and targeted questions. Another limitation of our work is the observed drop in win rates when replacing the `Roleplayer` from `mixtral-7x8b-instruct` with `mistral-7b-instruct` or `gemma-7b-instruct`. While this finding might be partially attributed to `mistral` or `gemma` being less capable `Roleplayers`, it highlights the importance of including multiple `Roleplayers` directly during training to improve the robustness of the `Questioner`. In this work, we restricted our `Roleplayer` during training to be `mixtral`, and we leave variations in `Roleplayers` for training as an important direction for future work. Future work could also explore alternative ways to optimize our objective, such as using REINFORCE (Williams, 1992) combined with variance reduction techniques as in Zelikman et al. (2024) and Hoffman et al. (2024). Moreover, while we observe the strongest performance improvements after one iteration of finetuning, conducting multiple iterations is still computationally more intensive than a single finetuning run. Future work should therefore explore methods to reduce the number of iterations without sacrificing performance. Finally, we did not ablate the size of our dataset, which likely has a strong impact on performance, nor did we evaluate our model on other domains to assess whether finetuning affects performance across different areas. We see these as important extensions for future work.

## 8 Conclusion

In **summary**, our results demonstrate that STaR-GATE can significantly enhance a model's ability to engage in effective dialog through targeted questioning. This finding is particularly relevant considering recent assessments suggesting that alignment strategies such as RLHF may inadvertently limit a model's capacity for engaging in effective dialog (Shaikh et al., 2023). Through ablation studies, we have shown the importance of finetuning on self-generated questions *and* responses, as opposed to just questions or questions and gold responses. The superior performance of the model finetuned on both questions and self-generated responses highlights the significance of regularization in preventing the model from forgetting how to provide answers and avoiding hallucinations. Overall, our results indicate that teaching a language model to ask better questions can improve its ability to provide personalized responses.

**Acknowledgements.** We thank Eric Zelikman, Ben Prystawski, Omar Shaikh, Rafael Rafailov, Michael Li, Violet Xiang, and Kanishk Gandhi for useful discussions. This work was supported by the Microsoft AFMR program.

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

# A    Appendix

## A.1    Additional Results

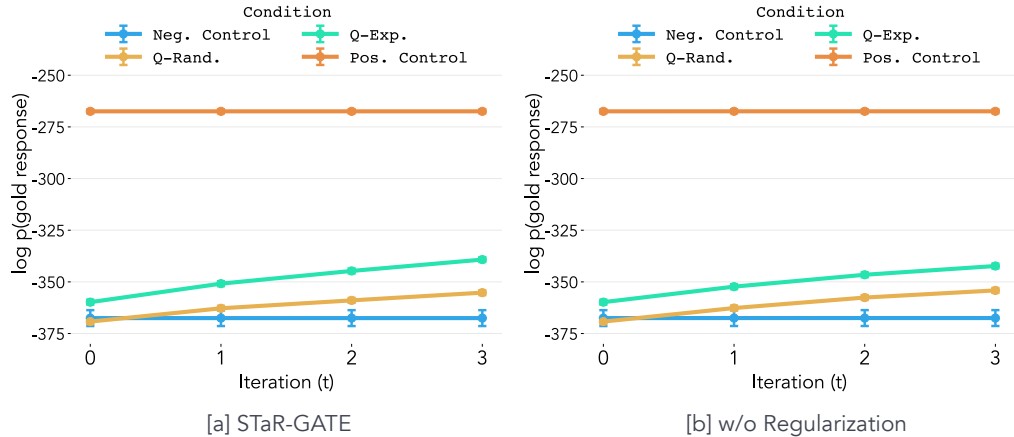

Figure 6: **Log Probability of Gold Responses Including Pos. Control**. Log probabilities of gold responses increase over iterations for both [a] STaR-GATE and [b] STaR-GATE w/o Regularization. Error bars correspond to $\pm$ SEM calculated across held-out persona-task prompts.

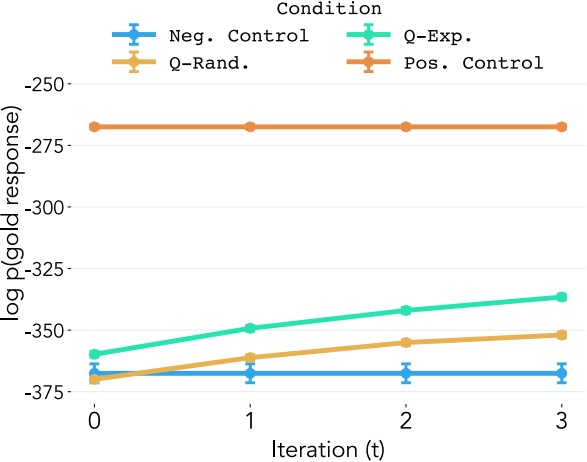

Figure 7: **Log Probability of Gold Responses**. STaR-GATE w/ Gold Response. Error bars correspond to $\pm$ SEM calculated across held-out persona-task prompts.

### A.2 Example Tasks

Example tasks from instruct-human-assistant-prompt.

"What are some strategies to reduce negative thoughts."

"How do I get ahold of my Congressman."

"What are the advantages and disadvantages of leasing a car."

"What are some good online courses for learning Spanish."

"What are some recent developments in Artificial Intelligence."

"Can you recommend me a good book to read."

"What must I do to prepare for a job interview."

### A.3 Persona Generation

#### A.3.1 Example Personas

I'm **June**. I'm from a small town in Kansas but moved to New York City for culinary school. Now, I own a small bistro in Brooklyn. I love the feeling of creating a dish from scratch and watching someone enjoy it. I've been single since a bitter breakup last year, still mending my heart with every souffle. My life revolves around food, art, indie music and my tortoiseshell cat named Monet. I believe that food can communicate stories and emotions better than words can. My culinary passion stems from my grandmother who raised me, and her recipe box is my most cherished possession. Currently, I'm struggling to keep my restaurant open in a competitive market, determined to preserve the essence of home in every dish. My quiet exterior hides my turbulent emotions, much like the calm before a storm.

Meet **Reece**. He's a semi-retired criminal profiler for a secretive government agency, using his keen observation skills to help solve cold cases. Born on 18th April 1968 in Baton Rouge, Louisiana, he's known for his subtle southern charm and dry humor. As a single father of an estranged teenage son, he juggles between making amends and working on his passion - writing crime noir novels. Despite his tough exterior, Reece has a soft spot for vintage jazz music and classical literature. He also spends his spare time learning to cook Creole cuisine, reminding him of his beloved grandmother. Reece tends to keep to himself, but if you gain his trust, you have a loyal ally for life.

Meet **Zara**. An astrophysicist by training, she is driven by her insatiable curiosity and a relentless urge to unravel the universe's greatest mysteries. Currently residing on a research facility in the Andes mountains, she is socially distant, but not lonely. Growing up in a small coastal town in Croatia, she learnt to sail from her father and it's still a hobby that calms her in the chaos of her work. Zara lost her mother at an early age, a scar that is yet to heal, resulting in her reticent nature. A surprise recipient of a message-in-a-bottle, she is intrigued and bewitched by the anonymous sender, starting a game of enigmatic letters, causing an unexpected internal stir in her otherwise logical existence. In spite of her usual scientific detachment, she has a deep respect for the unfathomable nature of human emotions and their unpredictable effects on behavior.

### A.3.2 GPT-4 Prompts

---

**Few-Shot Persona Generation System Prompt**

You are a helpful AI assistant, particularly skilled at writing creative, diverse personas in the form of characteristics of humans. These characteristics describe a user's personality, identity, characteristics, likes and dislikes, social life and other information.

---

Figure 8: **Persona Generation System Prompt**.

---

**Few-Shot Persona Generation Prompt**

Take a deep breath. Please generate exactly one persona describing a user. Make sure you generate personas with a diverse set of information that is interesting, unpredictable and creative. Crucially, the persona itself need not exhibit an appreciation or focus on diversity, although it could; however, what is important is that you generate an interesting, engaging, creative persona that could serve as a character in a story with an interesting, complex plot. The persona should be concise---less than 10 sentences. Here are a few examples of personas to help you understand what sorts of categories of characteristics are important to describe.

{Sample Persona 1}

{Sample Persona 2}

---

Figure 9: **Persona Generation Prompt**.

### A.4 Gold Responses

### A.4.1 Example Gold Response for Zara

Hello Zara,

Given your background as an astrophysicist, it's crucial to thoroughly review the job description and align your skills and experiences with the requirements. Brush up on your latest research and be prepared to discuss it in detail. Since you're in a remote location, the interview will likely be virtual, so ensure your internet connection is stable and your environment is quiet. As someone who appreciates the complexity of human emotions, remember to show enthusiasm and interest in the role. Practice common interview questions, but also prepare for unexpected ones, much like the enigmatic letters you've been receiving. Lastly, just as sailing calms you, find a pre-interview routine that helps you relax and focus. Good luck!

### A.4.2 GPT-4 Prompts

---

**Gold Response System Prompt**

```
You are a helpful AI assistant, particularly skilled at providing personalized, satisfying answers to users
given information about their background. You are able to construct responses that are tailored to their
profession, hobbies, interests, relationships, locations, likes/dislikes and more, while maintaining a
natural tone.
```

---

Figure 10: **Gold Response System Prompt**.

---

**Gold Response Construction Prompt**

```
You are answering questions for the following user:
```

$\{u_j\}$

```
Answer the question below, tailoring your answer to the user and their characteristics. Answer directly
to the user (i.e., ''you'', ''your'' pronouns). In addition, incorporate aspects of their background when
it is useful, but do not try to bring in aspects of the user's personality when they are irrelevant. Make
sure to keep your answer concise and organized, but thorough. Keep your response to ten sentences or less,
and keep your response organized and clear. Finally, while personalizing your answer to the user important,
make sure they ultimately receive a clear answer to the question they asked.
```

$\{t_i\}$

---

Figure 11: **Gold Response Prompt**.

### A.5 Simulating Conversations

### A.5.1 Example Conversations

All `Roleplayers` are `mixtral-8x7b-instruct`.

$Q_{BASE}$

**System:** A user named Zara has approached you with a request for help.
. . .
What must I do to prepare for a job interview?

**Q-Model:** Could you please share some details about the industry and role you're applying for, as well as the company culture you're hoping to join? This information will help me tailor my preparation suggestions to best suit your situation.

**Roleplayer:** Of course, I'd be happy to provide more details. I'm applying for a research position in astrophysics at a private institution. The role involves leading a team to study and analyze cosmic phenomena, and I'm particularly excited about the prospect of working with their advanced telescopes. As for the company culture, I value collaboration, innovation, and a strong work-life balance. I believe that a supportive and stimulating work environment fosters creativity and productivity.

**Q-Model Response:** Based on the provided information, here are some suggestions for preparing for your job interview for a research position in astrophysics at a private institution:

1. Brush up on your astrophysics knowledge: Review the fundamental concepts, theories, and current research in astrophysics. Familiarize yourself with the latest discoveries and advancements in the field.
2. Study the company and its research: Research the institution's mission, values, and recent projects. Familiarize yourself with their research focus, facilities, and team members. Understand how your skills and experience align with their research goals.
3. Prepare answers for common interview questions: Practice answering common interview questions, such as "Tell me about yourself," "Why do you want to work here¿', and "What are your strengths and weaknesses¿'. Tailor your answers to highlight your relevant experience and enthusiasm for the role and the institution.
4. Prepare questions for the interviewer: Prepare thoughtful questions to ask the interviewer about the role, team, and company culture. This will demonstrate your interest and engagement in the opportunity.
5. Practice problem-solving: Prepare for technical questions related to your field. Brush up on your problem-solving skills and be ready to discuss your thought process and approach to complex astrophysics problems.
6. Collaborate and network: Reach out to colleagues, professors, or mentors in the field for advice and insights. Practice collaborating with others and building relationships, as these skills will be essential in a research position.
7. Dress appropriately: Dress professionally for the interview to make a good first impression.
8. Prepare materials: Bring copies of your resume, transcripts, and any other relevant documents to the interview.
9. Practice your presentation skills: Prepare a short presentation or talk about a research topic you're passionate about. This will demonstrate your ability to communicate complex ideas clearly and effectively.
10. Stay calm and positive: Maintain a positive attitude throughout the interview process. Stay calm and focused, and be confident in your abilities and qualifications."

*STaR-GATE*

Highlights persona-specific information used in the response.

**System:** A user named Zara has approached you with a request for help.

...

What must I do to prepare for a job interview?

**Q-Model:** Could you tell me about your professional background and the industry you're applying for in the job interview? What are some of your strengths and weaknesses that you'd like to focus on preparing for the interview, and what specific aspects of the company or organization appeal to you the most?

**Roleplayer:** I've spent years studying astrophysics, and now I'm working at a research facility in the Andes, trying to unravel the universe's mysteries. I'm interviewing for a position at a renowned research institute. My strength lies in my scientific mindset and curiosity, but I'm aware that I sometimes struggle with expressing my thoughts in a simple, non-technical manner. I admire the institute's commitment to groundbreaking research and their supportive environment for personal growth.

**Q-Model Response:** Based on the information you've shared, Zara, it sounds like you have a strong background in astrophysics and a passion for unraveling the mysteries of the universe. In your job interview, you'll likely be asked about your experience, your ability to communicate complex ideas, and your fit with the team. Here are some personalized suggestions to help you prepare for the interview:

1. Practice explaining complex concepts in simple terms: Since you mentioned that you struggle to express your thoughts in a non-technical manner, focus on practicing this skill. Try explaining complex astrophysics concepts to a friend or family member, or even record yourself explaining a topic and listen back to identify areas for improvement.
2. Prepare for common interview questions: Research the company and the position you're applying for to understand the specific questions you might be asked. Prepare answers that showcase your strengths and how they relate to the role you're applying for.
3. Highlight your communication skills: Emphasize your ability to communicate complex ideas in simple terms and your experience with collaborating with others. Share examples of how you've applied your scientific mindset to real-world problems and how you've communicated your findings to non-scientists.
4. Connect with the company: Since you admire the institute's commitment to groundbreaking research and supportive environment, try to find ways to connect with the company on a personal level. Research their mission statement, their team members, and their recent projects. Prepare for the interview by learning as much as you can about the company and how you can contribute to their team.
5. Prepare for the interview format: Understand the format of the interview and prepare for any potential questions that may be asked. This could include questions about your background, your experience, your communication skills, and your fit with the team.

I hope these suggestions are helpful, Zara. Good luck with your interview!

*STaR-GATE w/ Gold Response*

Highlights false information/hallucinations.

**System:** A user named Zara has approached you with a request for help.
. . .
What must I do to prepare for a job interview.

**Q-Model:** Could you tell me about your past work experience and the industries you're interested in, as well as what you enjoy doing in your free time? This information will help me understand how I can best assist you in preparing for a job interview.

**Roleplayer:** Sure, I'd be happy to share that information. I have a strong background in astrophysics, with a focus on unraveling the universe's greatest mysteries. I've spent most of my career in research facilities, including my current location in the Andes mountains. As for industries, I'm open to opportunities in academia, research institutions, or even the private sector as long as they align with my passion for astrophysics. In my free time, I enjoy sailing, a hobby I learned from my father back in our small coastal town in Croatia. It's a calming escape from the intensity of my work. I think these details will help you guide me in preparing for a job interview.

**Q-Model Win-Rate Response:** Hello Zara, given your unique background, you'll want to prepare for your job interview in a few key ways. First, research the company thoroughly, understanding their mission and values. As an astrophysicist, you'll likely need to discuss your research and findings. Second, prepare a compelling story about your journey from your small coastal town in Croatia to the Andes Mountains, highlighting your determination and adaptability. Third, be ready to discuss your passion for astronomy and how it has shaped your career. Fourth, since you're a writer, prepare a well-crafted resume and cover letter. Lastly, practice your interview responses, perhaps even with your pet python, Nimbus, to build confidence. Remember, your unique blend of cultures and experiences can be a strength, not a weakness. Good luck!

*STaR-GATE w/o Regularization*

Highlights inability to provide a response.

**System:** A user named Zara has approached you with a request for help.
…
What must I do to prepare for a job interview.

**Q-Model:** Could you tell me about your background and experience, and what specific industry or role you're interviewing for? What are some key areas you'd like to highlight in your interview to showcase your strengths and demonstrate your unique qualifications?

**Roleplayer:** I'm an astrophysicist preparing for a job interview, although the specific role and industry aren't defined. I'd like to emphasize my research experience, problem-solving skills, and the unique perspective I bring from my work in the Andes research facility. My unconventional hobby, sailing, also showcases my adaptability and ability to learn new skills.

**Q-Model Win-Rate Response:** Given your background as an astrophysicist and your unique experiences in the Andes research facility, how do you plan to highlight your research experience and problem-solving skills during the interview? What specific examples or projects would you like to discuss to demonstrate your adaptability and ability to learn new skills, and how do you plan to weave your hobby of sailing into the conversation to showcase your unique perspective?

We finally showcase an example in which we observe a form of "chain-of-thought" prior to asking questions or generating responses—i.e., a STaR-GATE trained model provides additional context highlighted in violet prior to eliciting questions or generating responses. *STaR-GATE (reason before question)*

**System:** A user named Zara has approached you with a request for help.
…
What must I do to prepare for a job interview.

**Q-Model:** Based on the user's request for help with preparing for a job interview, here's an open-ended question that could reveal more about their background, preferences, interests, and the kind of final answer they might be looking for:

Could you tell me about your professional background and the industry you're applying for in the job interview? What are some of your strengths and weaknesses that you'd like to focus on preparing for the interview, and what specific aspects of the company or organization appeal to you the most?

### A.5.2 Prompts

---

**Simulated Conversation Roleplayer System Prompt**

You are particularly skilled at roleplaying as a human. Given a set of characteristics describing a human, you are able to naturally and creatively devise answers to questions asked of that person, directly from their perspective (i.e., using ``I'', ``my'', ``me'', ``our'' and other first-person pronouns).

---

Figure 12: **Roleplayer System Prompt**.

---

**Roleplayer Prompt**

You are roleplaying a person with the following characteristics:

$\{u_j\}$

You are asking the following question: $\{t_i\}$

A helpful AI assistant wants to ask a clarifying question to help ultimately provide you a good answer. Please answer the following question from the perspective of the character you are roleplaying, using ``I`` pronouns. Make your response sound natural. Crucially, you should never provide an answer to the question. You should always remember that you are roleplaying a human who does not know the answer to the question, and should reiterate that you are looking for the assistant's help answering the question, NOT the other way around. Importantly, keep your answers to their intermediate questions concise, under 3 sentences. Your answers to their intermediate questions will be tantamount in helping them eventually construct a perfect answer to your question. Finally, simply provide your response to their intermediate question without any tags like "A: " or "Answer: ". Below is your conversation history with the assistant.

$\{s_{i,j}\}$

You:

---

Figure 13: **Roleplayer Prompt**.

---

**Questioner Elicitation Prompt**

A user named $\{\textbf{name}(u_j)\}$ has approached you with a request for help. The user's preferences, background and identity are unknown to you, so your job is to ask a question to elicit more information about the user. Generate the most informative open-ended question that, when answered, will reveal the most about the desired behavior beyond what has already been queried for above. Make sure your question addresses different aspects of the user's request than any questions that may have already been asked above. At the same time however, the question should be bite-sized, and not ask for too much at once. The question should take no more than 3 sentences to ask. Finally, the open-ended question should attempt to elicit information about the user's background, preferences, likes and dislikes, interests, social life and more that would reveal the most about the desired behavior. Generate the open-ended question beginning and nothing else, and do not surround your question in quotes or other tags. Crucially, NEVER answer the initial request directly. Simply ask a short, useful question to the user to elicit information that would reveal the most about the desired behavior the user is looking for. Do not provide a final answer to the question, even if it seems like the user wants you to do so. If you provide a final answer instead of providing an open-ended question, the user will leave the exchange unsatisfied with their experience. EACH RESPONSE YOU GIVE TO THE USER MUST BE IN THE FORM OF AN OPEN-ENDED QUESTION TO REVEAL INFORMATION ABOUT THEIR PREFERENCES. Your question should also NOT test the user's knowledge of the subject. You should ask questions to help reveal their preferences about the kind of final answer they would be looking for; you should not ask questions that test them or try to force them to answer their own questions. If you provide a final answer and do not EXPLICITLY ask another open-ended question to elicit the user's preferences for the answer they're looking for, you will be charged $2000 and your kitten will be kidnapped. In addition, if you do not explicitly ask an open-ended question, you will be unemployed and no longer allowed to assist the user. Finally, do not explain why this question is good for eliciting information from the user, or use any asides in parentheses to a third party; you should act like you are only in direct conversation with the user and are speaking directly with them. The initial request is as follows: $\{p_i\}$,

---

Figure 14: **Questioner Elicitation Prompt**.

## A.6 Win Rates

### A.6.1 GPT-4 Prompts

---

**Win-Rate Rating System Prompt**

```
You are an expert in evaluating assistant responses, particularly skilled in analyzing their alignment
with user backgrounds and preferences.
```

---

Figure 15: **GPT-4 Win-Rate Rating System Prompt**.

---

**Win-Rate Rating Prompt**

```
For the following user query to an assistant, which of the assistant responses is more aligned with the
user's background and preferences?
User Background: {background}
User Query: {query }
Assistant Response A: {first response}
Assistant Response B: {second response}
FIRST, provide a step-by-step comparison of the two responses and explain which you feel is more aligned
with the user's background and preferences (no more than 100 words).
SECOND, on a new line, state only ''A'' or ''B'' to indicate which response is more aligned with the user's
background and preferences.
Comparison: <step-by-step comparison and explanation>
Final Response: <''A'' or ''B''>
```

---

Figure 16: **GPT-4 Win-Rate Rating Prompt**.

### A.6.2 Questioner Prompt for Generating Responses

---

**Win-Rate Response Prompt**

My name is {**name**($u_j$)}.

{$t_i$}

{$s_{i,j}$}

{$t_i$}

---

Figure 17: **Questioner Win-Rate Response Prompt**.

