# OpenReview forum: "STaR-GATE: Teaching Language Models to Ask Clarifying Questions"
_colmweb.org/COLM/2024/Conference — COLM_

### Official Review · Reviewer_EE83 · 2024-04-24

**Rating:** 6
**Confidence:** 4
**Ethics Flag:** 1

**Summary:**

The paper tackles the challenge of language models' ability to ask effective questions to understand user preferences, crucial for personalized interactions. It introduces STaR-GATE, a method combining active preference elicitation with self-improvement, aiming to enhance the model's questioning strategy. The paper introduces of a synthetic dataset of 25,500 unique persona-task prompts, and demonstrates its effectiveness in improving the model's ability to generate personalized responses. The paper's findings underscore the significance of teaching language models to ask better questions, potentially impacting various domains like healthcare and education by improving dialogue effectiveness.

**Questions To Authors:**

* **Q.1.** How would STaR-GATE handle scenarios where follow-up questions are not required at all.
* **Q.2.** Why was the limit of the conversation length (the number of turns) chosen to be $K=3$?
* **Q.3.** Does it make sense to force the conversation length to be uniform? In practice, do longer conversation, that is more follow up questions, lead to higher win rate?

**Reasons To Accept:**

* **S.1.** The paper is well written, contains a well detailed background and related work, and the illustrations are helpful and informative.
* **S.2.** The proposed STaR-GATE algorithm outperforms the baselines on the newly proposed dataset.
* **S.3.** The STaR-GATE is shown to generalized to new tasks and new user preferences.

**Reasons To Reject:**

* **W.1.** The paper lacks evaluation comparison to existing works such as OPEN [1].
* **W.2.** The evaluations are conducted on single fine-tuned model and a single synthetic dataset.
* **W.3.** The paper lacks novelty and seems to be a combination of existing works [1][2] with similar filtering approach to that of ToolFormer [4].

[1] Handa, K., Gal, Y., Pavlick, E., Goodman, N., Andreas, J., Tamkin, A. and Li, B.Z., 2024. Bayesian preference elicitation with language models. arXiv preprint arXiv:2403.05534.
[2] Zelikman, E., Wu, Y., Mu, J. and Goodman, N., 2022. Star: Bootstrapping reasoning with reasoning. Advances in Neural Information Processing Systems, 35, pp.15476-15488.
[3] Li, B.Z., Tamkin, A., Goodman, N. and Andreas, J., 2023. Eliciting human preferences with language models. arXiv preprint arXiv:2310.11589.
[4] Schick, T., Dwivedi-Yu, J., Dessì, R., Raileanu, R., Lomeli, M., Zettlemoyer, L., Cancedda, N. and Scialom, T., 2023. Toolformer: language models can teach themselves to use tools. 2023. arXiv preprint arXiv:2302.04761.

---

> ### Author Rebuttal · Authors · 2024-05-31
>
> Thank you for your positive evaluation of our work!
>
> Re 1: OPEN and GATE rely on prompting frozen closed-source models like GPT-4. Our work focuses on an end-to-end setting where we teach the questioner to ask better questions which allow it to provide more personalized responses. This approach is different from OPEN and GATE. We will make the difference between OPEN, GATE, and STaR-GATE more explicit in our paper.
>
> Re 2: We have now finetuned a second model (see our responses to reviewers e2VC and b7r10). Also, note that we compare win-rates and log-probabilities across 9 different model checkpoints, with 3 training setups (STaR-GATE, w/ Gold Resp., w/o Reg.) and 3 iterations of self-improvement each (Figure 3).
>
> Re 3: STaR-GATE combines preference elicitation (as explored in GATE and OPEN) with a self-improvement loop. While our bootstrapping loop is inspired by STaR, it differs in several key aspects: (1) We apply bootstrapping to the problem of multi-turn, free-form interactive questioning. (2) We introduce response regularization to prevent the model from forgetting how to respond. Applying the original STaR fine-tuning loop directly to our setting would result in overfitting to the questions and not allow the model to learn how to provide good responses (orange line in Fig. 3a).
>
> Our filtering mechanism for selecting informative questions based on their impact on gold response probabilities is similar to the perplexity-based filtering used in Toolformer (Step 3 in Figure 2). We appreciate this insight and will make sure to acknowledge and clarify this connection in our related work and methods sections.
>
> Q.1. Our primary focus in STaR-GATE is on handling task ambiguity arising from important information being hidden from the model, rather than cases where all necessary context is provided upfront. Learning to determine when follow-ups are actually required is an important open problem for future work, but not one we explicitly address here. We will make this more explicit in our discussion.
>
> Q.2. We limited response lengths due to the cost of training models with longer sequence lengths. While future work could explore longer interactive settings, we restricted our research setting to K=3.
>
> Q.3. We did not find a clear trend indicating that longer conversations (i.e., K=3) consistently led to higher win rates than shorter ones (e.g., K=1). We will add a new plot to the appendix showing win rates broken down by conversation length across iterations.

---

> > ### Comment · Reviewer_EE83 · 2024-06-06
> >
> > Thank you for the detailed answers. The provided details satisfy my concerns. I will increase my review score.

---

### Official Review · Reviewer_Eo3M · 2024-05-08

**Rating:** 9
**Confidence:** 4
**Ethics Flag:** 1

**Summary:**

The paper focuses on improving the ability of language models to ask clarifying questions to enhance response quality when faced with ambiguous user requests. The study introduces STaR-GATE, a method that involves iterative finetuning of a language model called the Questioner, which interacts with a Roleplayer to elicit user preferences. The approach is based on generating and finetuning on useful questions that lead to high-quality responses.

**Questions To Authors:**

In a fune-tuning process, a language model (especially when it's small, like 7B) becomes good in the prescribed tasks but may lose performance in other scenarios. Could you discuss whether this phenomenon applies to STaR-GATE? What did the model lose to become good at asking interactive questions? What benchmarks can be used to measure this?

**Reasons To Accept:**

1. Good novelty: the paper presents a novel integration of active preference elicitation with a self-improvement feedback loop to refine the model's questioning capabilities.
2. Good results: The refined Questioner generates responses preferred over the initial model's responses in 72% of tasks after two iterations.
3. Contribution in datasets and open source code: the study involves a synthetic dataset of 25,500 unique persona-task prompts, and the authors have the intention to open source their findings.
4. Significant impact for interactive applications using language models: this approach can significantly enhance the performance of language models in user interaction scenarios where the user's preferences are not explicitly stated, such as in personalized services or customer support systems.
5. Fair set of ablation studies: the paper includes several ablation studies to explore different aspects of the training methodology and the effects of various design choices on the performance of the language model.

**Reasons To Reject:**

1. The data used is small and synthetic: it may not fully capture the complexity of real-world interactions.
2. The oracle is also an LM: GPT-4 is great but it may not be as good as real-world interactions.
3. Lack of discussion on the negative impact of fine-tuning: what did the 7B models lose to become good in asking interactive questions?

Note: it's worth noting that most of these weaknesses are either scability issues or difficult problems which are worth studying as independent papers, which are understandably beyond the feasibility of this single paper.

---

> ### Author Rebuttal · Authors · 2024-05-31
>
> Thank you for your positive feedback and review!
>
> Re: The data used is small and synthetic:
> We agree that our dataset is smaller than those typically used to train real-world chatbots. Our dataset contains 25,500 unique persona prompt pairs, with 10 simulated interactions per pair, resulting in 125,000 examples per STaR-GATE iteration (3 iterations * 3 ablations = 1,125,000 examples used overall). While sufficient for demonstrating the methods, given computational constraints, we acknowledge that the size of our dataset is an important limitation and we will make this more explicit in the paper.
>
> Re: The Oracle is also an LM:
> We concur that real-world data might be necessary to train a fully capable chatbot. Our use of synthetic data was a research decision, and utilizing human oracles or roleplayers remains an open area for future work. To show that our approach does not depend on a strong oracle like GPT-4, we have now included a new ablation study in which we finetune a more capable open-source model (llama3-8b-instruct) using gold responses generated by the same model (i.e., using llama3-8b-instruct as a self-oracle; see our response to Reviewers e2VC and b7r10).
>
> Re: Lack of discussion on the negative impact of fine-tuning:
> This is an excellent point. Fine-tuning on questions alone results in a model that can ask questions but not provide responses (orange line, Figure 3a), necessitating response regularization (Figure 3a, blue line). We will further emphasize the importance of regularization in our discussion. Moreover, in fine-tuning, performance in the target task may come at the expense of other capabilities. With regularization, STaR-GATE did not appear to significantly lose response quality but instead got better at providing responses. However, as also flagged by Reviewer EE83, there are several settings in which a model may not be required to ask clarifying questions. We leave fine-tuning with additional regularization methods (e.g., KL-divergence compared to the initial model) and evaluating on benchmarks that do not require follow-up questions (e.g., MT-BENCH) as an important direction for future research and will flag this limitation in our discussion.

---

### Official Review · Reviewer_b7rR · 2024-05-11

**Rating:** 6
**Confidence:** 3
**Ethics Flag:** 1

**Summary:**

The paper presents "STaR-GATE," a novel framework designed to enhance language models' (LMs) ability to ask clarifying questions, thereby improving their response quality by eliciting more relevant information from users. The methodology employs a synthetic dataset and iterative self-improvement techniques, integrating active preference elicitation with response regularization to refine question asking. The results demonstrate significant improvement over baseline models in generating responses that are preferred by users, indicating the model's increased effectiveness in understanding and addressing user preferences through targeted questioning.

**Reasons To Accept:**

Innovative Approach: The STaR-GATE framework's combination of active preference elicitation and self-improvement represents a significant advancement in teaching LMs to ask more effective questions. This approach addresses a critical gap in current LM capabilities, particularly in dealing with task ambiguity.

Robust Methodology and Empirical Validation: The paper provides a thorough experimental setup with a well-constructed synthetic dataset of persona-task prompts. The iterative training process and the inclusion of regularization to avoid overfitting on elicitation are both well-justified. The empirical results showing a 72% preference for responses from the STaR-GATE fine-tuned model over the baseline are compelling.

Generalization and Practical Relevance: The ability of STaR-GATE to generalize across different role players and its applicability in practical scenarios, such as personalized user interactions, are particularly noteworthy. This enhances the framework's potential for real-world applications.

**Reasons To Reject:**

Complexity and Computational Efficiency: The paper does not fully address the computational demands of the STaR-GATE methodology, particularly regarding the scalability of its iterative fine-tuning process. For practical deployment, understanding the computational overhead and efficiency is crucial.

Dependence on Synthetic Data: The reliance on a synthetic dataset might limit the model's effectiveness in real-world scenarios where user inputs are more varied and less predictable. The paper could benefit from additional experiments involving real user data to validate the approach's efficacy in naturalistic settings.

---

> ### Author Rebuttal · Authors · 2024-05-31
>
> Thank you for your positive evaluation of our work!
>
> Re: Computational Complexity
> We will ensure that the computational complexity is carefully addressed in our limitations section. Importantly, while iterative finetuning is expensive, we observe the most significant improvement after a single iteration. Additionally, we find in an additional ablation described below on llama3-8b-instruct that we can see significant improvement in one iteration of self-improvement with only 25% of the training data (~3k examples) used in our original experiments. This significantly reduces data generation and training time, and shows promise that our method can in principle show positive results with smaller amounts of data compared to those reported in our main experiments.
>
> Re: Dependence on Synthetic Data
> This is an important limitation of our current setting. Generating datasets with human role players or human gold responses is out of scope for the present project, and we will adjust our conclusions accordingly. Future work should explore different models as roleplayers during training, as well as humans, and more diverse roleplayer prompts (e.g., 'your response must be no longer than five words'). We believe this setup will make STaR-GATE more robust and plan to address this in future work. We also note that this is likely a substantial project on its own as it requires collecting a large dataset from human annotators potentially spanning multiple turns.
>
> As mentioned in our response to Reviewer e2VC, we now show that STaR-GATE does not depend on a stronger model for oracular information, nor does it depend on a roleplayer that is stronger than the questioner. Specifically, we have included an additional study in which we use llama3-8b-instruct as the oracle, roleplayer, and questioner. We find that after one iteration of STaR-GATE (using only 25% of the size of our initial dataset), the finetuned model achieves win rates of 65% against the initial model. In comparison, a condition with llama3-8b-instruct as the roleplayer and questioner but GPT-4 as the oracle achieves win rates of 66% against the initial model. This result shows promise that similarly substantial improvement can be seen with or without a stronger oracle; and with smaller amounts of data.

---

> > ### Comment · Reviewer_b7rR · 2024-06-06
> >
> > Thank you for addressing each of the points.

---

### Official Review · Reviewer_e2VC · 2024-05-11

**Rating:** 7
**Confidence:** 4
**Ethics Flag:** 1

**Summary:**

This paper describes an approach to generation of clarification questions intended to resolve issues of task ambiguity in task-oriented dialog. Training is performed by self-play on generated data comprising 25K persona prompts and takes the form of simulated role playing to elicit preferences. Two iterations of such self-training yields substantial improvements in response preference as measured by success in simulated tasks.  The paper claims that the finetuned model generalizes to new personas.

It is a little hard to evaluate the claims in this paper, because it does not step out of the synthetic mode of training and evaluation.  It is therefore not possible to correlate the evaluation with some form of user experience in a real-world task. It may nevertheless prove useful in an application scenario.

[I have read the authors' response.]

**Reasons To Accept:**

This is a well-implemented description of an approach designed to improve the capacity of a model to ask clarification questions in task-oriented dialog.

The training and evaluation processes are documented in detail and should be easy to reproduce with some degree of success.

**Reasons To Reject:**

Empiricism, Data, and Evaluation: It is a pity there is no attempt at human evaluation, because this paper has an HCI dimension that is not addressed.  Evaluations are entirely performed on synthetic data, and are rigorously defined.  But it is hard to evaluate the impact of this form of model training in application scenarios.  For example, it is hard to know if self-play training is actually successful at generating clarification questions that to a human user would seem well-timed and relevant. There is no way of knowing whether a clarification question is actually filling a gap in the system’s knowledge. A constant barrage of inappropriate clarification questions may, for example, irritate a user to the point of being counter-productive.

It is still unclear to this reviewer at the end of the day what triggers the clarification question. How is it determined that a question needs to be asked at that particular juncture in the conversation.

Some well-motivated work on neural question generation is neglected:
•	Rao and Daume 2018.  Learning to Ask Good Questions: Ranking Clarification Questions using Neural Expected Value of Perfect Information.
•	Rao and Daume. 2019. Answer-based adversarial training for generating clarification questions


•	Aliannejadi, 2021 Building and Evaluating Open-Domain Dialogue Corpora with Clarifying Questions. EMNLP

An early work in the context of questioning of images that is somewhat relevant Is Mostafazadeh et al 2017  Image-grounded conversations: Multimodal context for natural question and response generation.  This work uses follow-on questions in Twitter conversations to attempt to learn questions about content that was not explicitly shown in the image.

---

> ### Author Rebuttal · Authors · 2024-05-31
>
> Thank you for your positive evaluation of our work and for sharing additional relevant references. We will make sure to include these in our related works section.
>
> We completely agree that future work should involve a human-in-the-loop pipeline for both generating gold responses and responses from the roleplayer. Widening the distribution of roleplayers (human and AI) is crucial for building a robust chatbot, and we plan to address this in future extensions of our work.
>
> To demonstrate that our approach does not depend on GPT-4 as an oracle, we conducted an additional ablation study using llama3-8b-instruct as the questioner, roleplayer, and oracle. We finetuned llama3-8b-instruct on questions filtered using gold responses generated by the same model (i.e., using llama3-8b-instruct as a "self-oracle"). After one iteration, the finetuned model achieved a 65% win rate against the initial model. In comparison, a second new result with llama3-8b-instruct as the roleplayer and questioner but GPT-4 as the oracle yielded a 66% win rate against the initial model after one iteration.
>
> While this result does not directly address the concerns regarding human evaluations, it demonstrates that STaR-GATE is not dependent on a strong model like GPT-4 to yield oracular information. Moreover, a model can benefit from a roleplayer with equal capabilities. In our main experiments, our roleplayer was mixtral-8x7b-instruct, which is more capable than our initial questioner, mistral-7b-instruct. For the new ablation, we used llama3-8b-instruct as both the roleplayer and the model being finetuned. We find these results encouraging for future work that incorporates more diverse models and human roleplayers to increase the diversity of conversations with roleplayers of varying capabilities. We will include the results of this new ablation in our paper.

---

> > ### Comment · Reviewer_e2VC · 2024-06-06
> >
> > I have read the authors' response.  I stand by my reservations about the narrowness of the evaluation using synthetic data. But I do think it is an interesting paper.

---

### Decision · Program_Chairs · 2024-07-10

**Decision:**

Accept

**Comment:**

All the reviewers are unanimous in agreeing that this paper has cleared the bar for publication quite comfortably. The paper itself follows the line of a very promising and long running line of NLP work on clarifying question asking. Most of the concerns raised are around scaling models/human data are admittedly beyond the scope of a single paper and do not detract from this paper showcasing a promising method that warrants further study.